# Visualization of Sirtuin 4 Distribution between Mitochondria and the Nucleus, Based on Bimolecular Fluorescence Self-Complementation

**DOI:** 10.3390/cells8121583

**Published:** 2019-12-06

**Authors:** Jeta Ramadani-Muja, Benjamin Gottschalk, Katharina Pfeil, Sandra Burgstaller, Thomas Rauter, Helmut Bischof, Markus Waldeck-Weiermair, Heiko Bugger, Wolfgang F. Graier, Roland Malli

**Affiliations:** 1Gottfried Schatz Research Center, Chair of Molecular Biology and Biochemistry, Medical University of Graz, Neue Stiftingtalstraße 6/6, 8010 Graz, Austria; jeta.ramadani@medunigraz.at (J.R.-M.); benjamin.gottschalk@medunigraz.at (B.G.); sandra.burgstaller@medunigraz.at (S.B.); thomas.rauter@medunigraz.at (T.R.); helmut.bischof@medunigraz.at (H.B.); markus.weiermair@medunigraz.at (M.W.-W.); wolfgang.graier@medunigraz.at (W.F.G.); 2Division of Cardiology, Medical University of Graz, 8010 Graz, Austria; katharina.pfeil@medunigraz.at (K.P.); heiko.bugger@medunigraz.at (H.B.); 3BioTechMed Graz, Mozartgasse 12/II, 8010 Graz, Austria

**Keywords:** array confocal laser scanning microscopy, fluorescence microscopy, fluorescent protein, genetically encoded sensor, mitochondrial protein import, self-complementing split FP technology, sfGFP, sfCherry2, sirtuin 4, structural illumination microscopy

## Abstract

Mitochondrial sirtuins (Sirts) control important cellular processes related to stress. Despite their regulatory importance, however, the dynamics and subcellular distributions of Sirts remain debatable. Here, we investigate the subcellular localization of sirtuin 4 (Sirt4), a sirtuin variant with a mitochondrial targeting sequence (MTS), by expressing Sirt4 fused to the superfolder green fluorescent protein (Sirt4-sfGFP) in HeLa and pancreatic β-cells. Super resolution fluorescence microscopy revealed the trapping of Sirt4-sfGFP to the outer mitochondrial membrane (OMM), possibly due to slow mitochondrial import kinetics. In many cells, Sirt4-sfGFP was also present within the cytosol and nucleus. Moreover, the expression of Sirt4-sfGFP induced mitochondrial swelling in HeLa cells. In order to bypass these effects, we applied the self-complementing split fluorescent protein (FP) technology and developed mito-STAR (mitochondrial sirtuin 4 tripartite abundance reporter), a tripartite probe for the visualization of Sirt4 distribution between mitochondria and the nucleus in single cells. The application of mito-STAR proved the importation of Sirt4 into the mitochondrial matrix and demonstrated its localization in the nucleus under mitochondrial stress conditions. Moreover, our findings highlight that the self-complementation of split FP is a powerful technique to study protein import efficiency in distinct cellular organelles.

## 1. Introduction

The sirtuin (Sirt) isoforms Sirt3, Sirt4, and Sirt5 harbor an N-terminal mitochondrial targeting sequence (MTS), responsible for the localization of these enzymes in the lumen of mitochondria [1]. Several studies, however, have reported that these Sirts can also occur outside of mitochondria [2,3,4], while the molecular mechanisms responsible for the subcellular distribution of Sirts are largely unclear. Sirt4 is an adenosine diphosphate (ADP)-ribosyltransferase enzyme located in the mitochondrial matrix [1,5] that regulates the metabolic activities of the organelle [6,7,8,9], but might also influence processes within the nucleus [2]. The subcellular distribution of Sirt4 is still debated [2], as well as in the case of Sirt3 [3,4,10,11]. Some studies not only indicate their localization within mitochondria, but also report translocations between the nucleus and mitochondria upon cellular stress [3,12]. Therefore, Sirt3 and Sirt4 potentially contribute to the mitochondrial unfolded protein response (mtUPR) as part of retrograde mitochondria-to-nucleus crosstalk. [13,14,15]. The mtUPR aims to restore mitochondrial protein homeostasis upon mitochondrial stress or perturbations [16,17,18,19,20]. The complete picture and overall roles of Sirt4 and the other mitochondrial Sirts have remained enigmatic.

Mitochondria comprise about 1500 proteins, of which the vast majority are nuclear encoded and hence translated by cytosolic ribosomes [21]. Distinct pathways of mitochondrial protein import (MPI) machinery control the targeting of these proteins to different sub-compartments of the organelle [21,22,23,24]. Mitochondrial matrix proteins (MMPs) harbor an N-terminal MTS [25,26,27], which is recognized by components of the translocase of the outer mitochondrial membrane (TOM) and the translocase of the inner mitochondrial membrane (TIM), catalyzing the import of MMPs [28,29]. During the import of MMPs across mitochondrial membranes, the nascent polypeptides remain unfolded. In mammalian systems, it is not yet entirely clear whether cytosolic and mitochondrial chaperones maintain the unfolding of these proteins, which are essential for the efficient importation of MMPs. There is some evidence that MMPs are cotranslationally imported into mitochondria and do not require additional chaperones [30]. Under stressed conditions, the capacity of the mitochondrial protein import machinery can be exceeded, [10,31] leading to an abundance of MMPs outside of the mitochondria. As a result, some of these proteins transfer signals of mitochondrial stress response to other cellular compartments, predominately to the nucleus, if they harbor, in addition to the MTS, a nuclear localization sequence (NLS) [32,33]. Such dually targeted proteins, e.g., ATF5, represent important components of the retrograde signaling pathways that significantly contribute to the mtUPR [34]. The identification and characterization of central players in the mtUPR is a subject of ongoing research. Although, the involvement of Sirts, a class of enzymes that regulate posttranslational modifications, in cellular stress responses has been described [35,36], however, their particular role in mtUPR remains poorly understood. In mammals, Sirts are classified into seven isoforms, referred to as Sirt1–7, exhibiting either mono-ADP-ribosyltransferase or deacetylase activity [37,38,39,40]. Besides controlling mammalian energy metabolism [41,42,43], Sirts are also involved in aging [44,45] and age related diseases [40,46,47], tumorigenesis [48,49], gene expression [50,51], and oxidative stress response [52,53,54].

Reliable methods that enable the investigation of the subcellular protein transport, distribution, and translocation of Sirts in intact living cells are essential to further reveal their involvement in distinct cellular stress responses. Most studies that have aimed to reveal the subcellular distribution of Sirts have applied methods such as immunohistochemistry to label (over)expressed Sirts fused with respective tags [1,2,10,55,56,57]. This approach requires fixation, and hence does not allow the investigation of the distributions and translocations of Sirts in living cells. Other invasive methods to study Sirt subcellular distributions have included the Western blotting of cell fractions [55,56,58]. In some works [1,5,6,59,60], Sirts, including Sirt4, have been fused to green fluorescent protein (GFP) variants or small non-fluorescent tags to study their subcellular localization on the single cell level. Some studies generated stably expressing Sirt4-GFP cell lines utilizing human fibroblasts for the analysis of subcellular Sirt4 abundance. These reports demonstrated that Sirt4-EGFP, as well as Sirt4-FLAG, colocalize with mitochondrial markers [1,5,6,59,60]. However, these studies are less informative in terms of the existence of extra-mitochondrial Sirt4.

Here, we fused Sirt4 with a very fast-maturing FP variant, the superfolder GFP (sfGFP) [61], which enables not only localization analysis as an endpoint-readout, but also the dynamic visualization of de novo protein synthesis [62]. Interestingly, Sirt4-sfGFP accumulated within the outer mitochondrial membrane (OMM) without entering the mitochondrial matrix. However, the fast folding sfGFP, which fused to a well-established MTS, was efficiently imported into the lumen of mitochondria. These findings point to a slower import efficiency of Sirt4 into mitochondria. Moreover, the accumulation of Sirt4-sfGFP within the OMM was associated with significant mitochondrial swelling, indicating a toxic effect of this particular fusion construct. Another strategy to investigate the importation of proteins into the mitochondrial matrix is based on split FP technology [63]. Bimolecular fluorescence self-complementation has already been successfully applied to visualize the importation of aggregated proteins into mitochondria in yeast [64]. To bypass the mislocalization of bulky Sirt4-FP fusion constructs and mitochondrial swelling, we took advantage of the split FP technology, based on the self-complementation of sfCherry2 [65,66]. Dividing the full FP sfCherry2 between the tenth and eleventh β-strands, the resulting sfCherry2_11_ fragment is a short peptide, consisting of only 17 amino acids (AAs) [65], which we fused C-terminally to Sirt4. In addition, we equally co-expressed the complementary sfCherry2_1–10_ constructs targeted to both the lumen of mitochondria and the nucleus, respectively. We named the novel tripartite probe mito-STAR, which means, in long form, mitochondrial sirtuin 4 tripartite abundance reporter. Our data demonstrate that the tripartite probe is suitable for the visualization of imported Sirt4 in both the mitochondrial matrix and nucleus upon the induction of mtUPR.

## 2. Materials and Methods

### 2.1. Buffers and Solutions

Cell culture materials were purchased from Greiner Bio-One (Kremsmünster, Austria). For plasmid expression, chemically competent 10-beta *Escherichia coli* (*E. coli*) cells were obtained from New England Biolabs (Ipswich, MA, USA). Agar−Agar Kobe I, CaCl_2_, D-Glucose, HEPES, KCl, MgCl_2_, NaCl, NaOH, Trypton/Pepton, and yeast extract were purchased from Carl Roth (Graz, Austria). Agarose was purchased from VWR International (Vienna, Austria). Oligomycin A and antimycin were purchased from Sigma Aldrich (Vienna, Austria). The cells were imaged in a 2 Ca-buffer (2CaB) containing, in mM, 2 CaCl_2_, 140 NaCl, 5 KCl, 1 MgCl_2_, 1 HEPES, and 10 d-Glucose, where the pH was adjusted to 7.4 for all (all buffer salts were obtained from Roth, Graz, Austria).

### 2.2. Cloning

Cloning was performed according to standard cloning protocols provided by the manufacturer, and the final plasmid was verified by sequencing (Microsynth, Vienna, Austria). Primers for cloning were purchased from ThermoFisher Scientific (Vienna, Austria). PCR reactions were performed using Herculase II fusion DNA polymerase (Agilent, Santa Clara, CA, USA). Mt-sfGFP was achieved using an N-terminal tandem repeat of the COXVIII targeting sequence [67]. SfGFP-N1 was purchased from Addgene (plasmid number 54737) and was used for the isolation and PCR-amplification of sfGFP. To obtain mitochondrial sfGFP (mt-sfGFP), the MTS was N-terminally fused to the sfGFP via NheI and EcoRI in a mammalian expression vector pcDNA3.1 (−) (Invitrogen, Austria). For the cloning of mtDsRed, DsRed (plasmid number 11151) was purchased from Addgene and isolated via PCR-amplification. To obtain mtDsRed, the tandem repeat of the MTS (CoxVIII) was N-terminally fused to the DsRed [67]. Constructs based on the split FP technology and the Sirt4-sfGFP were purchased from Gene Universal Inc. (Newark, USA). The constructs were inserted in the mammalian expression vector pcDNA3.1 (−). The following constructs were obtained from Gene Universal Inc.: Sirt4-sfGFP, seCFP-omp25, MTS-sfCherry2_1–10_, Sirtuin 4-sfCherry2_11_, and MTS-sfCherry2_1–10_-SC-Sirt4-sfCherry2_11_-SC-sfCherry2_1–10_-NLS. MTS refers to the mitochondrial targeting sequence, using an N-terminal tandem repeat of the COXVIII; 2A peptide refers to the self-cleavage peptide; NLS refers to the indication of a 2A self-cleavage peptide; NLS refers to the nuclear localizing sequence as a repetition of three times [68].

### 2.3. Cell Culture and Transfection

HeLa cells were grown in DMEM (Sigma Aldrich), containing 10% FCS, 100 U·mL^−1^ penicillin, and 100 mg·mL^−1^ streptomycin INS-1 832/13 (INS-1). The cells were cultivated in Gibco RPMI 1640 media (ThermoFisher). The cells were transfected at a confluency of 60–70%, in 30 mm imaging dishes (for an array confocal laser scanning microscopy—ACLSM) and on 1.5 H high-precision glass cover slips (MarienfeldSuperior, for SIM), using the transfection reagent PolyJet (SigmaGen Laboratories, Rockville, MD, USA) 24 h prior to measurement. PolyJet was incubated for 24 h and the cells were transfected with 1 µg of DNA. All cells were maintained in a humidified incubator (37 °C, 5% CO_2_, 95% air). All experiments were performed 24 h after transfection.

### 2.4. Mitochondria Staining

For colocalization analysis, the cells were first washed once with a loading-buffer containing, in mM, 2 CaCl_2_, 135 NaCl, 5 KCl, 1 MgCl_2_, 1 HEPES, 2.6 NaHCO_3_, 0.44 KH_2_PO_4_, 0.34 Na_2_HPO_4_, 10 d-glucose (Roth), 0.1% vitamins, 0.2% essential amino acids, and 1% penicillin/streptomycin (Gibco), all at pH 7.4. The cells were then incubated in a loading buffer containing either 0.2 µM MitoTracker^TM^ Green FM (MTG) or 0.2 µM MitoTracker^TM^ Red FM (MTR) for 20 min. Afterwards, the cells were washed three times with a loading buffer and imaged in 2CaB. MTR was imaged with an excitation of 561 nm, and emission was captured at 644 nm. MTG was excited at 488 nm and the emission was captured at 516 nm. Tetramethylrhodamine methyl ester (TMRM; catalog number T668; Invitrogen) loading was done likewise for 40 min with a TMRM concentration of 50 nM. TMRM was excited at 550 nm and emissions were caught at 600 nm. TMRM remained on the cells during measurement.

### 2.5. Submitochondrial Localization of Sirt4-sfGFP and mt-sfGFP

The cells were transfected with Sirt4-sfGFP, mt-sfGFP, seCFP-OMP25, MTS-sfCherry2_1–10_, Sirt4-sfCherry2_11_, mito-STAR, or mCherry-TOM22, and stained with MTG, MTR, or TMRM, respectively, and imaged using SIM microscopy or ACLSM, respectively. For colocalization studies, the ImageJ plugin ‘coloc2’ was used to measure the Pearson correlation coefficient.

### 2.6. Depolarization of Mitochondrial Membrane Potential

For depolarization, the mitochondria cells were treated with 10 µM of oligomycin A and 5 µM of antimycin A. Incubation of the cells with oligomycin A and antimycin A was done 8 h after transfection and lasted overnight prior to measurements.

### 2.7. Construction of Structural Models of the Probes

3D analysis and models of all probes were predicted with the online tool Phyre2 [69] (Protein Homology/analogy Recognition Engine V 2.0). Analyses of the predicted proteins were performed using DeepView/Swiss Pdb viewer V4.1.0 (ExPASy, Swiss Institute for Bioinformatics (SIB), Lausanne, Switzerland).

### 2.8. Confocal Imaging

Twenty-four hours after transfection, the cells were imaged using an array confocal laser scanning microscope (ACLSM), based on a Zeiss Observer Z.1 inverted microscope, equipped with a Yokogawa CSU-X1 Nipkow spinning disk system, a piezoelectric z-axis motorized stage (CRWG3-200; Nippon Thompson Co., Ltd., Tokyo, Japan), and a CoolSNAP HQ2 CCD Camera (Photometrics Tucson, Arizona, USA). The probes were excited at 488 nm and 561 nm, respectively and emission was captured at 516 nm and 644 nm, respectively, using a 100× objective. Data acquisition and control were done using the VisiView Premier Acquisition software (2.0.8, Visitron Systems, Puchheim, Germany).

### 2.9. Super-Resolution Imaging

The SIM setup consisted of a 405-, 488-, 515-, 532- and a 561-nm excitation laser. A CFI SR Apochromat TIRF × 100-oil (NA 1.49) objective was mounted on a Nikon Structured Illumination Microscopy (N-SIM) System, with standard wide-field and SIM filter sets, and was equipped with two Andor iXon3 EMCCD cameras mounted to a two-camera imaging adapter (Nikon Austria, Vienna, Austria), in order to achieve super-resolution. For calibration and reconstruction of the SIM images, ImageJ (National Institutes of Health (NIH), Bethesda, MA, USA) and Metamorph (Molecular Devices) were applied. Laser adjustment was checked by projecting the laser beam through the objective at the top cover of the bright field arm of the microscope prior to each measurement. This process was automatically run by the NIS-Elements Advanced Research (AR) (4.51, Nikon Corporation, Tokio, Japan). The cells were transfected using PolyJet with the respective constructs and stained, if necessary, with MTG and/or MTR, respectively (Invitrogen, Thermo Fischer Scientific, Vienna, Austria).

### 2.10. Mitochondrial Morphology Assessment

For the evaluation of mitochondrial area (a), perimeter (*p*), minor (x), and major (y) axes of the mitochondria, the ImageJ particle analyzer was used. Aspect ratio (AR) was defined as $\mathrm{AR}=\frac{x}{y}$.

For the determination of the form factor (FF), the following formula was applied: $\mathrm{FF}=\frac{p^{2}}{4\pi a}$.

Image analysis was performed with the freeware program ImageJ.

### 2.11. Data Analysis and Statistical Analysis

The obtained data were analyzed using GraphPad Prism 5 Software (GraphPad Software, Inc., La Jolla, CA, USA) and Excel (2013, 15.0, Microsoft). For image analysis, MetaMorph (Molecular Devices) and the freeware program ImageJ (NIH, MA, USA) were applied. To calculate the ratio values of the nuclear fluorescence to the cytosolic fluorescence or the mitochondrial fluorescence, respectively, first, the ACLSM images were background subtracted on MetaMorph using a background region of interest (ROI). Each further step was performed with MetaMorph. For the determination of the cytosolic and the nuclear fluorescence, region measurement was performed by drawing a region randomly in the cytosol or by bordering the complete nucleus. If complete nucleus bordering was not possible, for instance, because of mitochondrial structures, a region measurement was done by randomly picking a position in the nucleus. For the evaluation of the mitochondrial fluorescence, an auto threshold for light objects in MetaMorph was used to set a mask. The threshold was manually set to encompass mitochondrial structures. Hence, the average mitochondrial fluorescence within the mask was determined. We applied GraphPad Prism 5 for the statistical analysis, using either an unpaired double-sided Student’s *t*-test or a one-way ANOVA with a Bonferroni post-hoc test. The number of independent experiments is indicated in the figure legends. The total number of measured cells is also indicated in the figure legends.

## 3. Results

### 3.1. Mitochondrial Targeting of Sirt4-sfGFP in HeLa and INS-1 cells Is Less Efficient Compared to mt-sfGFP

First, we investigated the subcellular localization of Sirt4-sfGFP (Appendix A) in living HeLa cells (Figure 1a,c,d,f) and the pancreatic β-cell line INS-1(Figure 1b,c,e,g), exploiting an array confocal laser scanning microscope (ACLSM). We used HeLa cells because there is evidence for the involvement of Sirt4 as a tumor suppressor in cancer [70]. INS-1 cells were exploited, since it has been discussed that Sirt4 modulates insulin secretion through the leucine metabolism [71]. Due to the fast maturation of sfGFP, this FP is preferably used as a fluorescent tag to study the trafficking and transport of nascent proteins [72]. We compared the subcellular distribution of Sirt4-sfGFP with mt-sfGFP (Appendix A), a fusion construct of sfGFP, with a 71 AA tandem repeat of the MTS of COXVIII (Appendix A), which represents an approved mitochondrial targeting signal [67]. Sixty percent of HeLa cells expressing Sirt4-sfGFP showed exclusive mitochondrial fluorescence (Figure 1a,c,f), confirming the mitochondrial localization of Sirt4, which harbors an N-terminal 28-AA-long MTS (Appendix A) [1]. However, the remaining 40% of HeLa cells expressing Sirt4-sfGFP showed clear fluorescence, arising from the cytosol and nucleus, in addition to the mitochondrial fluorescence (Figure 1a,c,f). Analyzing the ratio of fluorescence signals between the nucleus and the cytosol unveiled an equal distribution of Sirt4-sfGFP among these distinct compartments in the HeLa cells (Figure 1d). Despite a significant positive correlation between the expression levels and mitochondrial targeting efficiency (Figure 1f), even HeLa cells, expressing rather low levels of Sirt4-sfGFP, showed clear extra-mitochondrial signals (Ratio F_Nucleus_/F_Mitochondria_ ≥ 0.33, denoted by gray areas, which were defined as areas of significant extra-mitochondrial localization; Figure 1f). In contrast, all HeLa cells expressing mt-sfGFP displayed an exclusive mitochondrial fluorescence signal independent of the expression level of the mitochondria targeted FP variant (Figure 1c,f). These findings indicate that the Sirt4-sfGFP targeting efficiency to mitochondria in HeLa is rather weak. We obtained similar results using INS-1 cells (Figure 1b,c,e,g), of which almost 80% showed a clear extra-mitochondrial Sirt4-sfGFP signal (Figure 1c), including cells with low expression levels of the construct (Figure 1g). All INS-1 cells expressing various levels of mt-sfGFP showed a high mitochondrial targeting efficiency for the fluorescent construct (Figure 1c,g). These data confirm that the weak mitochondrial targeting of Sirt4-sfGFP compared to mt-sfGFP also is present in the pancreatic β-cell model.

### 3.2. Sirt4-sfGFP Locates Exclusively at the OMM and Induces Mitochondrial Swelling

Interestingly, applying high-resolution microscopy to HeLa cells expressing Sirt4-sfGFP unveiled typical tubular structures of mitochondria with a darker lumen surrounded by green fluorescence (Appendix A). We observed similar patterns of fluorescent mitochondria with a dark lumen upon expression of a super-enhanced cyan FP (seCFP) variant fused to an efficient OMM targeting peptide, derived from outer mitochondrial protein 25 (OMP25, Appendix A) [73]. Co-staining of cells expressing Sirt4-sfGFP with MitoTracker^TM^ Red FM, which integrates into the inner mitochondrial membrane (IMM) [74], resulted in low levels of colocalization between the green and red mitochondrial fluorescence, respectively (Appendix A). These data indicate the localization of Sirt4-sfGFP exclusively at the OMM but not in the mitochondrial matrix. Next, we performed a series of super-resolution imaging experiments using structural illumination microscopy (SIM) with intact HeLa cells to determine the sub-mitochondrial localization of Sirt4-sfGFP. SIM images and respective line scans revealed strong colocalizations between Sirt4-sfGFP and mCherry-TOM22, an approved marker of the OMM (Figure 2a,b). Through analyses of the cross-section intensity profiles of Sirt4-sfGFP and mCherry-TOM22 with subpixel accuracy using a double Gauss fit, we unveiled that both constructs overlap with a uncertainty of 12 nm (SD, *n * =  30/59 cells/mitochondria), indicating their colocalization to the OMM. Moreover, Sirt4-sfGFP neither merged with the red fluorescence of TMRM of the IMM (Figure 2c,d), nor a mitochondrial matrix targeted DsRed (mtDsRed, Figure 2e,f). These findings indicate that Sirt4-sfGFP is absent from the IMM and mitochondrial matrix in living cells, proving the exclusive localization of this fusion construct at the OMM. We further applied SIM to determine the sub-mitochondrial localization of mt-sfGFP in living HeLa cells, to exclude the possible influence of sfGFP on the mitochondrial import machinery. The co-expression of mt-sfGFP with mCherry-TOM22 showed no colocalization, while mtDsRed confirmed the matrix localization of mt-sfGFP by high colocalization (Appendix A).

Calculation of the form factor and aspect ratio, as the ratio between the major (height) and minor (width) axes of the ellipse equivalent to the mitochondrion, revealed that both the form factor and the minor of mitochondria in cells expressing Sirt4-sfGFP were significantly higher compared to cells expressing mt-sfGFP (Appendix A). This analysis points to swelling of mitochondria due to the expression of Sirt4-sfGFP. We did not find significant differences between cells expressing mt-sfGFP or Sirt4-sfGFP for the major axis and the form factor, indicating that the overall morphology of swelled mitochondria was not affected by Sirt4-sfGFP (Appendix A).

### 3.3. Self-Complementation of Split sfCherry2 Confirms the Localization of Sirt4 within the Mitochondrial Matrix

Our data so far emphasize that the fusion of the fast maturating sfGFP to the C-terminus of Sirt4 prevents the importation of the construct into the mitochondrial matrix. Consequently, Sirt4-sfGFP integrates into the OMM, which is associated with mitochondrial swelling. To overcome the mislocalization, we generated Sirt4-sfCherry2_11_, in which a 17-AA short β-barrel strand of sfCherry2 was fused to the C-terminus of Sirt4 (Appendix A), which was co-expressed with non-fluorescent mt-sfCherry2_1–10_ (Appendix A). In the case of an entire import of Sirt4-sfCherry2_11_ into the lumen of mitochondria, self-complementation with sfCherry2 is possible, yielding a detectable red fluorescence signal (Figure 3a). Using this established approach [64], we proved the importation and localization of Sirt4-sfCherry2_11_ to the mitochondrial matrix (Figure 3b). As expected, colocalization studies with either mt-sfGFP, MitoTracker^TM^ Green FM or seCFP-OMP25 confirmed the abundance of Sirt4-sfCherry2_11_ in complementation with mt-sfCherry2_1–10_ to exclusively reside in the lumen of mitochondria (Figure 3c–e).

### 3.4. Development of mitoSTAR, a Tripartite Self-Complementing FP-Based Probe for the Visualization of Mitochondria to Nucleus Distribution of Sirt4 in Living Cells

The successful self-complementing FP approach proved the intact and entire importation of Sirt4 into the mitochondrial lumen. This observation is well in line with several reports [6,59,60] demonstrating the localization and function of Sirt4 within mitochondria. Nevertheless, there is also evidence for the abundance of Sirt4 within the nucleus [2]. Hence, it appears relevant to investigate the possible distribution of Sirt4 between these two organelles. In order to visualize the import of Sirt4 into mitochondria and the nucleus, respectively, we extended the bimolecular probe with an additional nuclear-located non-fluorescent sfCherry2_1–10_ (sfCherry2_1–10_-NLS), using an approved nuclear localizing sequence (NLS) in a three-time repetition [68]. To avoid low efficient expression levels upon cotransfection and to achieve equal amounts of all three parts, i.e., MTS-sfCherry2_1–10_, Sirt4-sfCherry2_11_, and SfCherry21-10-NLS, we exploited the P2A peptide, which has a strong degree of self-cleavage activity (Appendix A) [75]. A schema of the respective plasmid coding for MTS-sfCherry2_1–10_, Sirt4-sfCherry2_11_, and SfCherry21-10-NLS which are separated by 2 self-cleavage sides, is shown in Appendix A. The self-complementation of the two non-fluorescent sfCherry2_1–10_ targeted to mitochondria and the nucleus, respectively, with Sirt4-sfCherry2_11_, generates a localized red fluorescence signal, which is a direct measure of the subcellular distribution of Sirt4 in one given cell (Figure 4a). We named this probe mito-STAR, meaning mitochondrial sirtuin 4 tripartite abundance reporter. The expression of mito-STAR in HeLa cells unveiled that in 80% of transfected cells, the self-complementation of sfCherry2 occurred exclusively within the mitochondrial matrix (Figure 4b,d,e), indicating the high importation efficiency of Sirt4-sfCherry2_11_ into mitochondria. Notably, HeLa cells expressing mito-STAR showed improved mitochondrial localization compared to HeLa cells expressing Sirt4-sfGFP (Figure 4d). However, in 20% of transfected HeLa cells, the self-complementation of sfCherry2 was also detectable within the nucleus (Figure 4b,d,e), indicating that in these cells Sirt4-sfCherry_11_ is additionally present within the nucleus. The overall fluorescence intensity, as a measure of the expression level of mito-STAR, did not positively correlate with the nuclear fluorescence in HeLa cells expressing the tripartite probe under control conditions (Pearson coefficient: 0.11). In 70% of INS-1 cells, we observed an exclusive mitochondria-located fluorescence signal, indicating an efficient import of Sirt4-sfCherry2_11_ into mitochondria (Figure 4c–e). However, the remaining 30% of transfected INS-1 cells also displayed clear nuclear red fluorescence, indicating the abundance of Sirt4 within the nucleus (Figure 4c–e).

Colocalization analysis of HeLa cells expressing mito-STAR, together with either Sirt4-sfGFP or mt-sfGFP, further demonstrated that Sirt4-sfGFP sticks at the OMM, while Sirt4-sfCherry2_11_ is colocalized with mt-sfGFP in the lumen of mitochondria (Appendix A). In contrast to Sirt4-sfGFP (Appendix A), the expression of mito-STAR in HeLa cells did not induce mitochondrial swelling (Appendix A).

We next utilized mito-STAR to visualize mitochondrial stress induced by cell treatment with oligomycin A and antimycin A, a combination of well-known mitochondrial toxins that efficiently depolarize the mitochondrial membrane potential and induce severe cell stress [76,77,78]. All HeLa cells treated with the toxins exhibited dominant nuclear red fluorescence, indicating a higher abundance of Sirt4-sfCherry2_11_ within the nucleus upon cell stress (Figure 4f,g).

## 4. Discussion

In this study, we aimed to visualize the subcellular distribution of Sirt4, a mitochondria targeted sirtuin, in living cells, by exploiting two different FP approaches. In a first attempt, we fused Sirt4 to sfGFP. Sirt4-sfGFP is supposed to become fluorescent within several minutes upon its biosynthesis, due to the fast maturation of sfGFP [79]. Hence, the imaging of cells that synthesize Sirt4-sfGFP enables acute visualization of the subcellular distribution of this construct under different conditions. In a second approach, we fused Sirt4 to a non-fluorescent 17-AA short peptide, which represents the 11th β-strand of the red fluorescent sfCherry2 [65]. Sirt4-sfCherry2_11_ requires self-complementation with the non-fluorescent sfCherry2_1–10_ to generate red fluorescence over time. Exploiting well approved targeting strategies for sfCherry2_1–10_, we used the self-complementation split FP technology for the visualization of the mitochondria-to-nucleus distribution of Sirt4-sfCherry_11_ [65].

In addition to Sirt4, Sirt3 and Sirt5 are also known to harbor an N-terminal MTS [10,80,81], a peptide with positively charged amino acids, essential for importation into the mitochondrial matrix [82]. However, the importation of proteins into mitochondria can stagnate under certain stress conditions, which consequently leads to an abundance of these proteins outside of the organelle [83,84]. If these proteins are not instantly degraded, they can initiate signaling pathways to further promote or counteract cellular stress [85]. Particularly, in the course of mtUPR, a stress condition due to the accumulation of unfolded proteins within mitochondria, the accumulation of mitochondrial targeted proteins within the cytosol and nucleus contributes to the retrograde signaling pathway [33,85]. While several reports point to Sirt3 being involved in mtUPR [13,14,86], less is known about the role of the other mitochondrial targeted Sirts, such as Sirt4 and Sirt5, in this particular stress response. During mitochondrial stress, Sirt3, with its histone deacetylase activity, emerges and accumulates within the nucleus, yielding an activation of FOXO3A [86]. In contrast to Sirt3 and Sirt5, Sirt4 exhibits very low or even no histone deacetylase activity [87], but instead other enzymatic activities, including the removal of lysine modifications, delipoylation, and ADP-ribosyltransferase activity [5,6,71]. There is evidence that these different enzymatic activities of Sirt4 modulate key metabolic enzymes in the lumen of mitochondria, thereby influencing insulin secretion, lipid metabolism, and tumorgenesis [5,48,70,71]. However, it remains elusive whether Sirt4 also contributes to the Sirt/mtUPR axis, and if the enzyme emerges within the nucleus under mitochondrial stress. While our data do not provide information about the enzymatic activity of Sirt4 fusion constructs, we demonstrate that in many cells these constructs also occur in the cytosol and nucleus upon its expression, indicating the existence of Sirt4 outside of mitochondria in addition. This assumption is further confirmed by our finding that Sirt4-sfGFP was not entirely imported into mitochondria and accumulated in the cytosol and nucleus, even at very low expression levels. In contrast to cells expressing Sirt4-sfGFP, cells expressing high levels of mt-sfGFP, in which the same FP variant was targeted to the mitochondrial matrix by an approved MTS, did not show any fluorescence arising from the cytosol and nucleus. Considering the equal translation and folding rates of both constructs, these data point to a slower and less effective import activity of Sirt4-sfGFP into mitochondria compared to mt-sfGFP. We found that the expression levels of Sirt4-sfGFP were comparable to those of mt-sfGFP. Hence, we assumed that the release of Sirt4-sfGFP from ribosomes and subsequent folding precedes its mitochondrial importation in many cells. The hypothesis of a slow mitochondrial import of Sirt-sfGFP was further confirmed by our finding that mitochondria located Sirt4-sfGFP sticks at the OMM. Additional experiments testing other FP variants with different folding kinetics are necessary to clarify if the rapidly maturing sfGFP prevents the complete import of the respective Sirt4 fusion construct into mitochondria. However, our results support the idea that due to highly susceptible mitochondrial Sirt4 import characteristics, this enzyme might represent an early and highly sensitive sensor of mtUPR, such as ATF5 [34].

Using high and super-resolution fluorescence microscopy, we have proven that mitochondria located Sirt4-sfGFP integrates into the OMM. This finding is unexpected, in light of the many other studies that have demonstrated the localization and function of overexpressed Sirt4 within the mitochondrial matrix [59,60,87]. In order to evaluate whether the OMM targeting of Sirt4-sfGFP was due to the bulky and fast maturating sfGFP [79], we exploited self-complementing split FP technology [65,88]. This sophisticated method has the massive advantage that a small 17-AA tag, fused to the protein of interest, is sufficient to visualize the subcellular distribution upon self-complementation with the remaining split FP fragment. Exploiting this technique allowed us to confirm the mitochondrial localization of Sirt4 to the mitochondrial matrix. To additionally visualize a possible abundance of Sirt4-sfCherry2_11_ in the nucleus, we expanded the probe by a nuclear-targeted complementary split FP. In this way we could detect the abundance of Sirt4-sfCherry2_11_ within mitochondria and the nucleus in the same cells, while the approach does not allow the detection of the construct within the cytosol. Although the mitochondrial targeting of Sirt4-sfCherry2_11_ compared to Sirt4-sfGFP was significantly improved, many cells showing a nuclear fluorescence signal were detectable, proving the nuclear abundance of Sirt4-sfCherry2_11_ under resting conditions. Applying the self-complementation-based tripartite Sirt4 abundance reporter (mito-STAR) under conditions of mitochondrial stress, this dramatically increased the Sirt4 level in the nucleus. These results again support the idea that Sirt4 contributes to mitochondria-to-nucleus retrograde signaling. Moreover, our data emphasize that this technique is useful to study the involvement of other proteins, such as Sirt3 and Sirt5, in the retrograde signaling pathway with high resolution. We also highlight that this approach is applicable to screen compounds that target mitochondrial protein import, such as stendomycin [31].

## 5. Conclusions

In summary, our work unveils that sfGFP fused to mitochondria targeted proteins, in combination with (super) high-resolution microscopy, could be exploited to identify mitochondrial stress responders exhibiting slow and poor mitochondrial importation activity. Moreover, we introduced a novel expandable tripartite probe, which bypasses the rather peculiar effects of sfGFP fusion, enabling the quantification of the distribution of a protein of interest between two distinct compartments by fluorescence microscopy.

## Figures and Tables

**Figure 1 cells-08-01583-f001:**
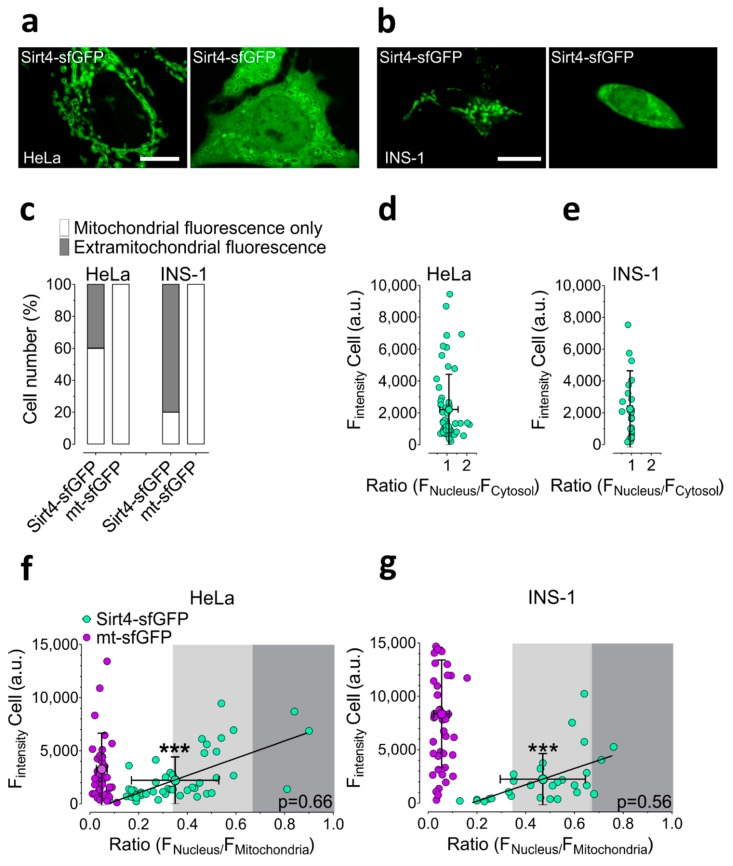
Subcellular distribution of sirtuin 4 (Sirt4)—superfolder green fluorescent protein (sfGFP) in HeLa and INS-1 cells. (**a**) Representative array confocal laser scanning microscopy (ACLSM) images of HeLa cells expressing Sirt4-sfGFP. Left image represents Sirt4-sfGFP located to mitochondria. Right image shows the extra mitochondrially located Sirt4-sfGFP. Scale bar indicates 10 µm. (**b**) Representative ACLSM images of INS-1 cells expressing Sirt4-sfGFP. Left image represents Sirt4-sfGFP targeted to mitochondria. Right image shows the extramitochondrially located Sirt4-sfGFP, scale bar indicating 10 µm. (**c**) Percentage of HeLa and INS-1 cells expressing either Sirt4-sfGFP or mt-sfGFP that displayed only mitochondrial fluorescence as opposed to extramitochondrial fluorescence; *n* = 97 cells from 3 independent experiments (HeLa); *n* = 73 cells from 3 independent experiments (INS-1). (**d**) Ratio of Sirt4-sfGFP nucleus fluorescence to Sirt4-sfGFP cytosolic fluorescence in HeLa cells plotted against whole cell fluorescence intensity. Data are shown as ratios with mean ± SD; *n* = 52 cells. Three independent experiments were done. (**e**) Ratio of nuclear fluorescence to cytosolic fluorescence of Sirt4-sfGFP in INS-1 cells plotted against the whole cell fluorescence intensity. Data are shown as ratios with the mean ± SD; *n* = 29 cells. Three independent experiments were performed. (**f**) Illustration of HeLa cells expressing either Sirt4-sfGFP or mt-sfGFP, calculated as ratio of nuclear fluorescence to mitochondrial fluorescence and plotted against whole cell fluorescence intensity. Correlation of Sirt4-sfGFP between F_Nucleus_ and F_Mitochondria_ is depicted as a mathematical function and evaluated with the Pearson coefficient shown in the lower right corner, indicated with “p”. Three independent experiments were performed for each condition. Data are shown as ratios with the mean ± SD; *n* = 97 cells. (*** *p* < 0.0001, unpaired double-sided t-test). (**g**) Calculation of the ratio of nuclear fluorescence to cytosolic fluorescence of INS-1 cells expressing either Sirt4-sfGFP or mt-sfGFP, plotted against the whole cell fluorescence intensity. Correlation of Sirt4-sfGFP between F_Nucleus_ and F_Mitochondria_ is illustrated as a mathematical function and evaluated with the Pearson coefficient in the lower right corner, indicated with “*p*”. Three independent experiments for each condition were performed. Data are shown as ratios with the mean ± SD; *n* = 73 cells (*** *p* < 0.0001, unpaired double-sided *t*-test).

**Figure 2 cells-08-01583-f002:**
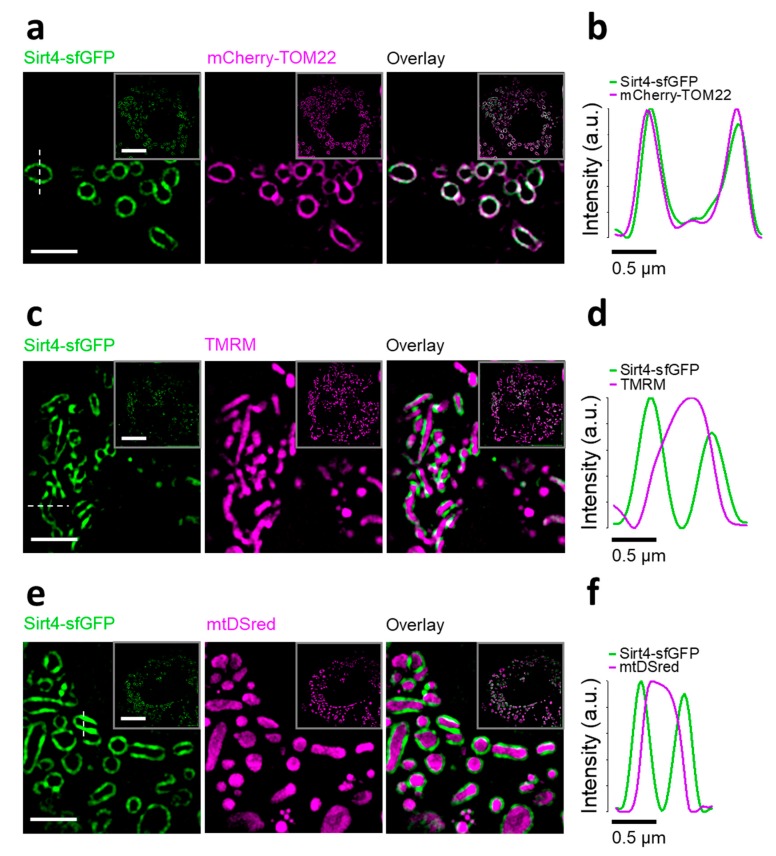
Sirt4-sfGFP is exclusively located at the outer mitochondrial membrane. (**a**) Representative SIM images of HeLa cells expressing Sirt4-sfGFP (green, left panel) and mCherry-translocase of the outer mitochondrial membrane 22 (TOM22, magenta, middle panel). Right panel displays overlay of Sirt4-sfGFP and mCherry-TOM22. Scale bar represents 2.5 µm. Squares in the upper right corner of each image show the whole cell. Scale bar represents 10 µm. Pearson correlation coefficient: 0.74 (mean) with a standard deviation of 0.10. (**b**) Representative line scan of an individual mitochondrion as demonstrated in (**a**) (left panel, white dashed line). Curves show respective fluorescence intensity of the line scan. (**c**) Identical experiments as those in (**a**), but using tetramethylrhodamine methyl ester (TMRM, magenta). Pearson correlation coefficient: 0.34 (mean) with a standard deviation of 0.14. *n* = 3. (**d**) Identical experiments as those in (**b**), but using TMRM (magenta). (**e**) Identical experiments as those in (**a**), but using mitochondrial matrix targeted DsRed (mtDsRed). Pearson correlation coefficient: 0.33 (mean) with a standard deviation of 0.07 (magenta). (**f**) Identical experiments as those in (**b**), using mtDsRed instead (magenta). Shown experiments are representative of 3 independent experiments, encompassing 30 different cells.

**Figure 3 cells-08-01583-f003:**
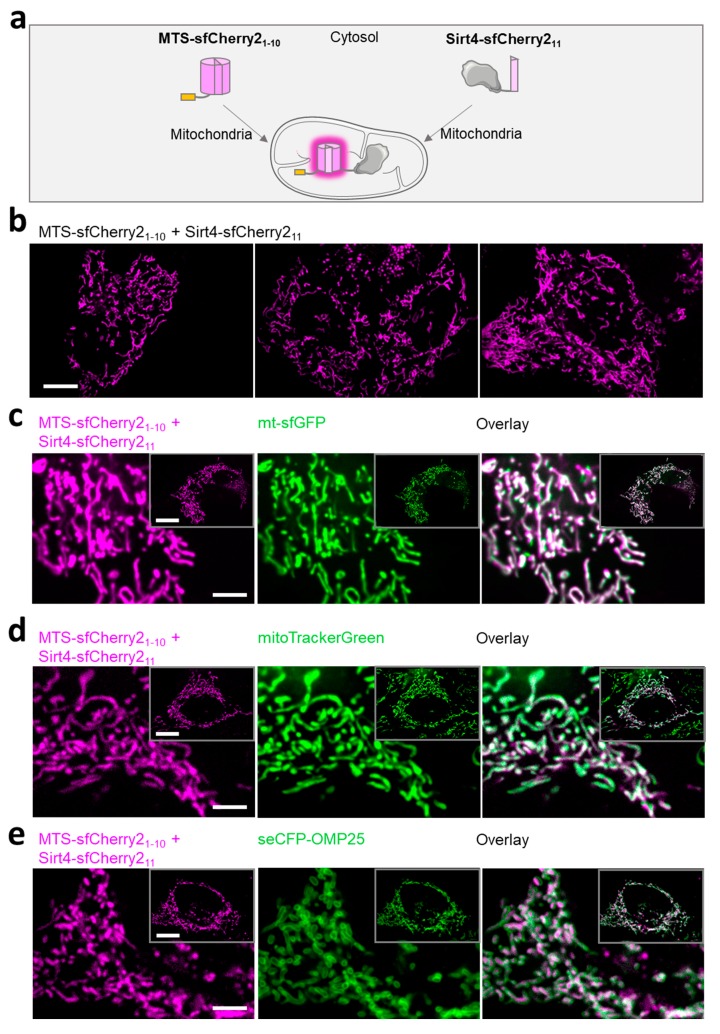
Application of the split FP variant, exploiting the self-complementation, proves the localization of sfCherry2 within the mitochondrial matrix. (**a**) Schematic overview of the split FP approach, consisting of the mitochondrial targeting sequence (MTS) fused to the sfCherry2_1–10_ and the Sirt4 fused to the corresponding sfCherry2_11_. Upon self-complementation of sfCherry2 in the lumen of mitochondria, a red fluorescence signal is detectable. (**b**) Three representative confocal images of HeLa cells, expressing both MTS-sfCherry2_1–10_ and Sirt4-sfCherry2_11_, depicted as fluorescence signals in magenta. Scale bar represents 10 µm. (**c**) Representative confocal images of HeLa cells expressing MTS-sfCherry2_1–10_ and Sirt4-sfCherry2_11_ (magenta) and mt-sfGFP (green). The overlain image is illustrated as a merge of both channels. Scale bar represents 10 µm. Squares in the upper right corner of each individual image of the series indicate the whole cell. Scale bar 10 µm. Pearson correlation coefficient: 0.78 (mean) with a standard deviation of 0.12. (**d**) Identical experiments as those in (**b**), but using MitoTracker^TM^ green FM (green). Pearson correlation coefficient: 0.73 (mean) with a standard deviation of 0.07 (**e**) Identical experiments as those in (**b**), but using seCFP-OMP25 (green). Pearson correlation coefficient: 0.64 (mean) with a standard deviation of 0.09. Shown experiments are representative of 3 independent experiments, encompassing 30 different cells.

**Figure 4 cells-08-01583-f004:**
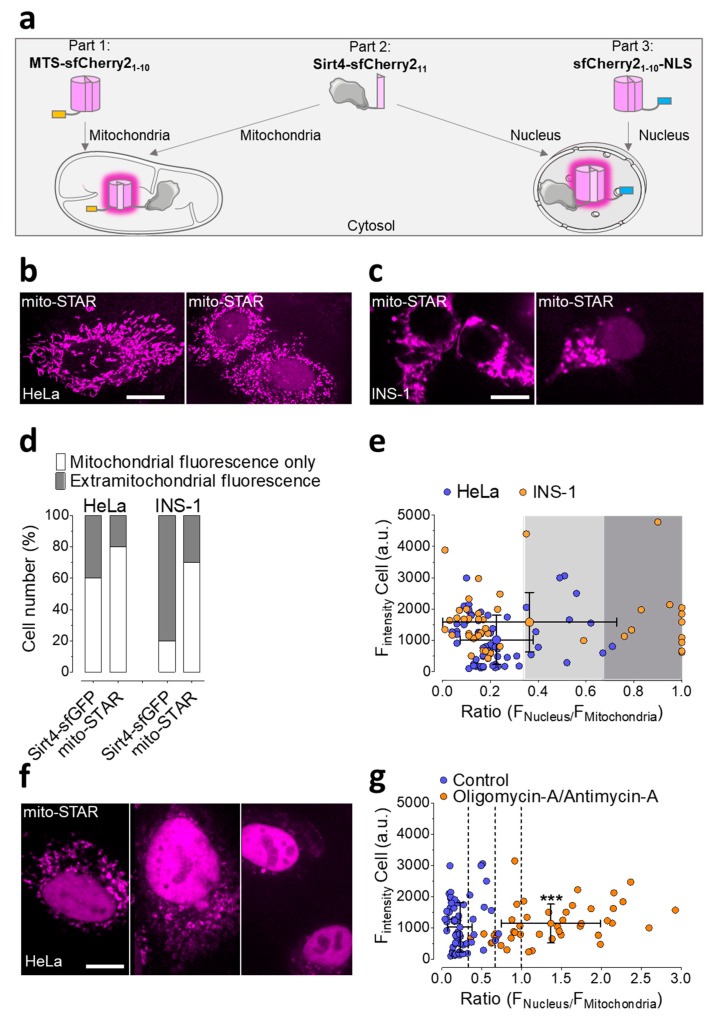
Design and application of the mitochondrial sirtuin 4 tripartite abundance reporter (mito-STAR). (**a**) Schematic illustration of the sensor principle. The importation of Sirt4-sfCherry2_11_ into mitochondria or the presence in the nucleus is shown as a red fluorescence signal (illustrated in magenta) in mitochondria and/or the nucleus, respectively. (**b**) Representative confocal images, showing HeLa cells expressing mito-STAR. Left image represents Sirt4-sfCherry2_11_ located to mitochondria only. Right image shows the extra mitochondrially located Sirt4- sfCherry2_11_ in addition to that located to mitochondria. Scale bar indicates 10 µm. (**c**) Representative confocal images of INS-1 cells expressing mito-STAR. Left image depicts Sirt4-sfCherry2_11_ located to mitochondria only. Right image shows the extra mitochondrially located Sirt4- sfCherry2_11_ in addition to that located to mitochondria. Scale bar indicates 10 µm. (**d**) Percentage of HeLa cells and INS-1 cells, expressing either Sirt4-sfGFP or mito-STAR, showing mitochondrial fluorescence only or extramitochondrial fluorescence. *N* = 120 cells (HeLa); *n* = 74 cells (INS-1). (**e**) Calculation of the ratio of nuclear fluorescence to cytosolic fluorescence of HeLa cells and INS-1 cells expressing mito-STAR, plotted against the whole cell fluorescence intensity. Data are shown as ratios with the mean ± SD; *n* = 67 cells (HeLa); *n* = 45 cells (INS-1); three independent experiments were carried out, respectively. (**f**) Three representative confocal images of HeLa cells expressing mito-STAR after 16 h treatment with 10 µM of oligomycin A and 5 µM of antimycin A. Scale bar indicates 10 µm. (**g**) Calculation of the ratio of nuclear fluorescence to cytosolic fluorescence of HeLa cells expressing mito-STAR after 16 h of oligomycin A (10 µM) and antimycin A (5 µM) treatment versus controls, plotted against the whole cell fluorescence intensity. Performance of 3 independent experiments for each condition. Dots are shown as ratio values ± SD. *N* = 111 cells from 3 independent experiments (*** *p* < 0.0001, unpaired double-sided t-test).

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
