# Peer review of "Visualization of Sirtuin 4 Distribution between Mitochondria and the Nucleus, Based on Bimolecular Fluorescence Self-Complementation"

_cells, 2019, doi:10.3390/cells8121583_

Round 1

Reviewer 1 Report

In the present manuscript the authors addressed a long sought after problem: cellular distribution of sirtuins, NAD+-dependent deacylases with a wide range of roles in energy metabolism modulation and transcription regulation. Using an elegant and at the same time sophisticated fluorescent sensor-based methodology that included a tripartite self-complementing FP-based probe, the authors unambiguously demonstrated mitochondrial matrix localization of Sirt4 and its mictochondria to nucleus distribution under mitochondrial stress conditions. The paper is very clearly written and the data support the conclusions. The only problem I noticed (although it could be a good problem to have) is a bit too lengthy Introduction. As such, I recommend acceptance of the current manuscript in the present form.

Author Response

Point 1: In the present manuscript the authors addressed a long sought after problem: cellular distribution of sirtuins, NAD+-dependent deacylases with a wide range of roles in energy metabolism modulation and transcription regulation. Using an elegant and at the same time sophisticated fluorescent sensor-based methodology that included a tripartite selfcomplementing FP-based probe, the authors unambiguously demonstrated mitochondrial matrix localization of Sirt4 and its mictochondria to nucleus distribution under mitochondrial stress conditions. The paper is very clearly written and the data support the conclusions. The only problem I noticed (although it could be a good problem to have) is a bit too lengthy Introduction. As such, I recommend acceptance of the current manuscript in the present form.

Response 1: We very much appreciate the positive evaluation of our manuscript by reviewer1. Indeed our introduction is quite long, while, to the best of our knowledge, it does not exceed the formal criteria of the journal. As indicated by reviewer 1 the advantage of our long introduction is that we could sufficiently refer to important aspects related to our work. Thus, we would prefer acceptance of the long version of our introduction.

Reviewer 2 Report

In this manuscript, the authors aim to investigate the subcellular localization of Sirt4 in mitochondria and nucleus through fluorescence imaging. They first fused Sirt4 with sfGFP due to its fast maturation. Their results show that the targeting efficiency to mitochondria of Sirt4-sfGFP is very weak and Sirt4-sfGFP only locate at the outer mitochondrial membrane (OMM) which causes the mitochondrial swelling. To overcome these issues, they further developed a mitochondrial Sirtuin4 Tripartite Abundance Reporter (mitoSTAR) based on the self-complementation split sfCherry2. This new probe allows them to successfully visualize the subcellular distribution of Sirt4. Overall, the experiments are carefully performed and the paper is well-written. Two small things to be considered in a revised version.

Line 231, should it be “Sirt4-sfGFP” rather than just “Sirt4” ? The last paragraph starting from line 459 should be classified to a “conclusion” section.

Author Response

In this manuscript, the authors aim to investigate the subcellular localization of Sirt4 in mitochondria and nucleus through fluorescence imaging. They first fused Sirt4 with sfGFP due to its fast maturation. Their results show that the targeting efficiency to mitochondria of Sirt4-sfGFP is very weak and Sirt4-sfGFP only locate at the outer mitochondrial membrane (OMM) which causes the mitochondrial swelling. To overcome these issues, they further developed a mitochondrial Sirtuin4 Tripartite Abundance Reporter (mitoSTAR) based on the  selfcomplementation split sfCherry2. This new probe allows them to successfully visualize the subcellular distribution of Sirt4. Overall, the experiments are carefully performed and the paper is well-written.

Two small things to be considered in a revised version.

Point 1: Line 231, should it be “Sirt4-sfGFP” rather than just “Sirt4” ?

Response 1: We thank reviewer 2 very much for the positive evalautation of our manuscript and the two insightful comments. Indeed it should be “Sirt4-sfGFP” in line 231. We have corrected the manuscript in this regard.

Point 2: The last paragraph starting from line 459 should be classified to a “conclusion” section.

Response 2: This is a nice suggestion. We have introduced a “conclusion” section. Thank you very much for this note.

Reviewer 3 Report

Ramadani-Muja et al., report the development of mito-STAR (mitochondrial sirtuin4 tripartite abundance reporter) for visualizing the localization of Sirt4 using super-resolution microscopy. This aims to provide an alternative tagging approach of using a split fluorescent protein (FP) and self-complementation in place of direct fusion of FPs to Sirt4 to avoid localization defects caused by the bulky fused FP. The authors demonstrated that using their approach, the proper localization of Sirt4 in the mitochondria can be visualized, as well as its transport to the nucleus during mitochondrial stress. This could potentially be applied for the screening of compounds that cause intra-organelle transport of proteins.

The manuscript is very well written and will be interesting for the readers of Cells. I recommend its publication in its current form with some minor changes outlined below.

Minor comments:

Label the co-localization (overlaid) images with pearson correlation values, wherever appropriate.

For image figures with multiple panels, it would be helpful to the reader if they are labeled more clearly. For example, Figure 1a has two panels shown side-by-side, what is the difference between these two panels? Is one expressing mt-sfGFP and the other Sirt4-sfGFP? One shows distinct punctuated fluorescence and the other shows more disperse signal across the entire cell. Figures should stand on their own and have appropriate information for interpretation. Same goes for Figures 1b, 3b, 4b, 4c, 4f.

Figure 4f – label each panel (control, treatment, etc.).

Author Response

Ramadani-Muja et al., report the development of mito-STAR (mitochondrial sirtuin4 tripartite abundance reporter) for visualizing the localization of Sirt4 using super-resolution microscopy. This aims to provide an alternative tagging approach of using a split fluorescent protein (FP) and self-complementation in place of direct fusion of FPs to Sirt4 to avoid localization defects caused by the bulky fused FP. The authors demonstrated that using their approach, the proper
localization of Sirt4 in the mitochondria can be visualized, as well as its transport to the nucleus during mitochondrial stress. This could potentially be applied for the screening of compounds that cause intra-organelle transport of proteins.
The manuscript is very well written and will be interesting for the readers of Cells. I recommend its publication in its current form with some minor changes outlined below.

Response: We thank reviewer 3 very much for the positive evaluation of our manuscript and the minor comments, which help to further increase the readability of the paper.

Minor comments:
Point 1: Label the co-localization (overlaid) images with pearson correlation values, wherever appropriate.

Response 1: We highly appreciate this suggestion. We tested inclusion of the pearson correlation values within the images themselves as suggested by reviewer 1. However, we feel that placing the values into or next to the images/figure led to an overload of information. Thus, we indicated the pearson correlation values of all overlaid images in the respective figure legends.

Point 2: For image figures with multiple panels, it would be helpful to the reader if they are labeled more clearly. For example, Figure 1a has two panels shown side-by-side, what is the difference between these two panels? Is one expressing mt-sfGFP and the other Sirt4-sfGFP? One shows distinct punctuated fluorescence and the other shows more disperse signal across the entire cell. Figures should stand on their own and have appropriate information for
interpretation. Same goes for Figures 1b, 3b, 4b, 4c, 4f.

Response 2: We understand this issue and corrected respective figures by adding white borders between respective separate images. In addition, we now explain respective images in more detail within respective figure legends.

Point 3: Figure 4f – label each panel (control, treatment, etc.). 

Response 3: Again, we felt that adding text to the figure 4f might be rather unfavourable. Hence, we explained respective conditions within the figure legend.
